# Needs Companion: A Novel Approach to Continuous User Needs Sensing Using Virtual Agents and Large Language Models

**DOI:** 10.3390/s24216814

**Published:** 2024-10-23

**Authors:** Takuya Nakata, Masahide Nakamura, Sinan Chen, Sachio Saiki

**Affiliations:** 1Graduate School of Engineering, Kobe University, 1-1 Rokkodai-cho, Nada-ku, Kobe 657-8501, Hyogo, Japan; 2The Center of Mathematical and Data Science, Kobe University, 1-1 Rokkodai-cho, Nada-ku, Kobe 657-8501, Hyogo, Japan; chensinan@gold.kobe-u.ac.jp; 3RIKEN Center for Advanced Intelligence Project, 1-4-1 Nihonbashi, Chuo-ku 103-0027, Tokyo, Japan; 4Department of Data & Innovation, Kochi University of Technology, 185 Miyanokuchi, Tosayamada-cho, Kami 782-8502, Kochi, Japan; saiki.sachio@kochi-tech.ac.jp

**Keywords:** need, service, virtual agent, large language model, personalization, human-centered design

## Abstract

In today’s world, services are essential in daily life, and identifying each person’s unique needs is key to creating a human-centered society. Traditional research has used machine learning to recommend services based on user behavior logs without directly detecting individual needs. This study introduces a system called Needs Companion, which automatically detects individual service needs, laying the groundwork for accurate needs sensing. The system defines a needs data model based on the 6W1H framework, uses virtual agents for needs elicitation, and applies large language models (LLMs) to analyze and automatically extract needs. Experiments showed that the system could detect needs accurately and quickly. This research provides interpretable data for personalized services and contributes to fields like machine learning, human-centered design, and requirements engineering.

## 1. Introduction

Industry 4.0, Industry 5.0, and Society 5.0 are key concepts for the next-generation information and communication technology (ICT) and artificial intelligence (AI) society, each addressing societal challenges from different perspectives [1,2,3]. All emphasize the importance of needs as a crucial keyword. Industry 4.0 focuses on improving efficiency and automating industrial processes to increase productivity and competitiveness [4]. In contrast, Industry 5.0 and Society 5.0 aim to use technology to build a human-centered society and improve the quality of life by addressing societal needs [5]. To achieve Society 5.0, a development ecosystem of services and systems finely tuned to various societal needs is necessary. In modern society, meeting individual needs through personalized services is essential [6]. The widespread use of smartphones has made service applications indispensable in daily life, and improving service quality directly enhances the quality of life [7]. However, there is insufficient research on concrete methods to detect individual service needs as data models. The fact that machine learning recommendations come before the crucial process of understanding personal needs is a significant issue [8]. This study revisits the fundamental question of what needs are and proposes a needs sensor capable of automatically detecting the service needs of individual users. It defines a data model for service needs and uses a virtual agent (VA) and large language model (LLM) for needs elicitation and analysis through voice dialogue [9]. Experiments were conducted to verify the sensor’s accuracy, resolution, and noise immunity. Software quality was also evaluated through surveys. Through this approach, the study establishes a technology capable of automatically sensing needs with sufficient accuracy.

This paper is structured as follows: Section 2 covers needs and personalization. Section 3 addresses issues in prior studies. Section 4 explains the objectives of this study. Section 5 describes the proposed method of the needs sensor. Section 6 details the implementation. Section 7 discusses the performance of the proposed method based on experiments. Finally, Section 8 provides a summary of this research.

## 2. Preliminaries

### 2.1. Need

In software development, it is crucial to identify what customers truly desire. This has been studied in fields such as requirements engineering and user-centered design [10,11]. Traditional methods include expert-led surveys and interviews, which are still widely used in development settings today [12]. Requirements are specific conditions or functions that guide system specifications and designs. In contrast, needs, which precede requirements, are more abstract problems or goals that users want to achieve [13]. Research on needs is well-established, drawing from theories such as Maslow’s hierarchy of needs and Deci’s self-determination theory [14,15]. Modern software services are essential in fields like healthcare, disaster prevention, marriage, employment, social networking, and entertainment. Service needs relate to all levels of Maslow’s hierarchy, from physiological needs to self-actualization. In self-determination theory, research on needs has focused on basic psychological needs (BPNs), which are essential for a psychologically healthy and fulfilling life [16]. BPNs are primarily classified into three categories: autonomy, competence, and relatedness. In services, BPNs should be seen not just as demands for functions but as essential elements of daily life that enhance happiness. Research shows that BPNs change with time, culture, life stage, and personal circumstances, and they do not always require conscious awareness [17,18]. To fulfill diverse needs through services, it is essential to accurately capture these changing needs. However, research on representing needs as data models is insufficient, and consequently, research on detecting the internal states of needs is also lacking.

### 2.2. Personalization

Personalization generally means “the act of making something suitable for the needs of a particular person” [19]. In digital technology, personalization has been extensively studied. It is defined as “the process of modifying a system’s functionality, interface, information access, content, and uniqueness to increase relevance to an individual or a group of individuals” [20]. A traditional approach to personalization involves users manually adjusting service settings, which is still prevalent in many applications today. This method allows users to directly fulfill their service needs. Recently, research on personalization using deep learning has become active in the field of machine learning [21]. The main method is collaborative filtering, where deep learning recommends appropriate services based on users’ usage history, such as subscriptions or product purchases [22]. Much research has also focused on personalization based on natural language dialogue between users and systems. One such method is the Conversational Search System [23]. This system sets a target during the initial conversation and then infers the desired service through rounds of questioning, responding, and searching, eventually presenting the result and gathering feedback on user satisfaction. There is also research on personalizing the agent that interacts with users [24]. In our laboratory, extensive research on personalization and agents is being conducted, with potential applications in conversational personalization. For example, we have studies on a smart system personalization framework [25] and a VA to support elderly conversations [9]. Conversational Recommender Systems (CRSs) also exist, using machine learning to process dialogue content [26,27]. These advanced machine learning-based recommendation systems enable rich interactions between users and the system, allowing users to provide questions and feedback.

## 3. Challenges of Previous Studies

Gathering service requirements necessitates specialists, making it impractical to frequently elicit individual requirements [28]. This approach indirectly extracts needs in the form of requirements essential for system design, leaving the true nature of the needs unknown. This study proposes a method to directly and automatically extracting needs without specialized knowledge. In studies using LLMs to automate requirements engineering, LLMs are used to summarize interview results or assist in interviews, but the interviews themselves are not automated [29]. In machine learning-based personalization, service usage data are used for training, raising privacy and security concerns, which can reduce users’ motivation [30,31]. Machine learning only recommends services based on data analysis, without directly interpreting needs, making it difficult to accurately understand the context of needs or the user intent [32]. This can lead to incorrect recommendations and a subsequent decrease in purchasing behavior. This study aims to obtain needs as structured data, thereby enhancing the correct understanding of needs and making them usable as highly explainable features. Dialogue with VAs is a useful method for detecting needs. Although VA interviews are common in areas like job interviews and counseling, they are rarely used to gather needs [33,34]. Research has focused on improving agent performance by using LLMs for tasks such as semantic interpretation, but using LLMs to structurally interpret needs is new [35]. Research on BPNs shows that needs-related experiences and behaviors become clear through natural language [36]. This study aims to improve personalization methods through dialogue, exploring designs that help extract needs.

## 4. Goal

The goal of this study is to propose, develop, and evaluate a system called Needs Companion that can automatically detect individual user service needs. Needs Companion functions as a type of needs sensor, and this study aims to experimentally clarify the performance of the proposed needs sensor. Evaluating the needs sensor, which detects human internal states, is challenging. Therefore, user-system interaction will be assessed based on response time, resolution, accuracy, noise immunity, and repeatability—common performance indicators used for sensors such as those measuring light and temperature.

## 5. Proposal Method

### 5.1. Key Idea

The core concept of Needs Companion is to automatically extract structured needs from spoken language using a VA for needs elicitation dialogue and an LLM for complex natural language analysis. The needs are structured into the 6W1H format. The approach of this study is as follows:(A1)6W1H needs data model;(A2)Needs extraction using LLM;(A3)Needs elicitation dialogue by virtual agent.

### 5.2. Architecture

The overall architecture of Needs Companion is shown in Figure 1. The user engages in one-on-one dialogue with the VA on the device. The user has specific needs related to a service. When the user verbally communicates these needs to the VA, the VA converts the speech into text using voice recognition. The LLM then processes the speech to obtain needs data expressed in the 6W1H format, thereby detecting the user’s needs. Based on the detected needs, the VA engages in further dialogue with the user, asking questions to achieve more accurate needs detection. At the end of the dialogue, the user is asked whether the needs were accurately detected, and feedback is collected.

### 5.3. (A1) 6W1H Needs Data Model

The proposed needs data model represents the three needs in BPN theory: autonomy, competence, and relatedness. For user needs related to services, it is important to determine what service they want to use (service) and what specific actions they want to perform (action). The motivation for the action, along with contextual factors such as time and place, represents autonomy. The action itself, which is the goal the user wants to achieve, represents competence. The relationships with others through the service represent relatedness. These can be expressed using the commonly applied 6W1H framework (how, why, when, where, what, who, whom). In this study, the service needs data model is defined using the 6W1H framework, as shown in Table 1.

### 5.4. (A2) Needs Extraction Using LLM

Using an LLM, 6W1H needs data are extracted from user utterances. The flow of needs extraction is shown in Figure 2. First, as a preparation for needs extraction, the system infers which service the user wants to use. This inference can be performed using a pre-prepared list of service names for string search or by generating service names with the LLM. During needs extraction, the user’s utterance and the inferred service name are input into the LLM. The prompt includes the content from Table 1 to explain the 6W1H needs data model, along with detailed extraction rules to refine the output. Specifically, the prompt instructs the system to always extract the how and what elements, which are foundational to the needs, and to prioritize the extraction of finer-grained elements like when and where elements over the more abstract why element. The extracted output is the 6W1H needs data. There are two important considerations for extraction. First, a single utterance may not refer to only one need; multiple service needs may be extracted simultaneously. Second, if the utterance does not mention a time reference, the ‘when’ element cannot be extracted, so it is not always possible to obtain all 6W1H elements in a single extraction. Therefore, a re-extraction mechanism is needed to update the previously extracted 6W1H needs data based on additional user input. During re-extraction, the system inputs both the new user data and the previously extracted 6W1H data into the LLM. By calculating the similarity of needs, the updated needs are identified from multiple needs outputs, maintaining the consistency of re-extraction.

### 5.5. (A3) Needs Elicitation Dialogue by Virtual Agent

Figure 1 shows the flow of needs elicitation through VA dialogue. The VA engages in dialogue with the user to elicit their needs. Based on research showing that needs-related experiences and behaviors become clear through language, this dialogue aims to surface needs [36]. The key steps of needs elicitation are clarification and final confirmation. If the extracted needs data are missing any 6W1H elements, the VA identifies this and asks the user for clarification to gather more detailed needs. The user, prompted by this request, reconsiders their needs and responds to the VA with any additional needs. While user awareness of needs is not essential for deriving satisfaction from fulfilling them, such awareness and externalization are necessary for detecting needs from the outside. Reconsideration of needs promotes this awareness. In the final confirmation, the VA asks the user directly if the detected 6W1H needs data are accurate. This step improves the accuracy of the needs detection and enhances the user’s sense of autonomy and competence, making them feel more satisfied when their needs are fulfilled [37,38].

## 6. Implementation

Figure 3 illustrates the implementation architecture. The VA, called Mei, runs on a PC browser [9]. Users interact via voice, and the system responds with voice, on-screen avatar gestures, expressions, and text. The system has a simple interface and responsive interactions from the VA, designed to be user-friendly even for older adults unfamiliar with digital devices. The dialogue screen is shown in Figure 4. The system uses the Google Speech API for voice recognition [39]. It is built on the Java virtual machine with three components, each having its own database: the Needs Extraction System, the User System, and the Service System. The Needs Extraction System has a needs database that stores 6W1H needs data. The User System manages user information, while the Service System allows pre-registration of service names and stores newly discovered service names from user utterances. The LLM runs through an LLM Wrapper implemented in Python 3.9, integrating prompts and settings to run the OpenAI API [40]. When the system was built in early 2024, OpenAI offered two models: the high-performance gpt-4 and the more affordable gpt-3.5-turbo. Given the high frequency of LLM usage in the proposed system, gpt-3.5-turbo is chosen for its practicality and cost-effectiveness. To ensure accurate needs extraction, the sampling temperature is set to 0.0. Few-shot prompting is utilized in the prompts, providing several examples of input and output [41]. In this implementation, four systems were constructed on the web, but flexible operations are possible, including handling personal information on an on-premise server or deploying these systems on a cloud using virtual servers. Cloud services like the Google Speech API and OpenAI API are used, leveraging cloud computing resources. However, the response time and control over output results using these APIs can be challenging.

## 7. Evaluations

### 7.1. Experimental Setup

To evaluate the sensor performance of Needs Companion, we conducted an experiment. Eight participants used Needs Companion continuously for five weeks. The group included six students and two faculty members from the laboratory. Three specific services—YouTube, smart home services, and social networking services (SNSs)—were designated as mandatory request tasks, with each participant required to submit at least one request for each. Additionally, participants were free to communicate requests for any other services. Prior to the experiment, participants received a brief tutorial on how to use Needs Companion. During the actual experiment, participants interacted with Needs Companion and conducted needs detection dialogues independently, without staff support. Participants could speak to Needs Companion at their convenience whenever they thought of a service need during the experiment period. The computer running Needs Companion had 8GB of memory and was a Surface Book with Windows 10 OS, using Google Chrome as the browser and built-in microphone and speakers for audio input and output. After the experiment concluded, participants completed a brief survey.

### 7.2. Experimental Results

We evaluated whether users judged their needs to be correctly detected during the final confirmation of the dialogue. This evaluation was based on whether the users responded “Yes” or “No” to the confirmation. A total of 88 dialogues were conducted across all participants. For each participant, the proportion of dialogues that continued until the final confirmation is summarized in Figure 5, and the proportion of “Yes” responses to the final confirmation is shown in Figure 6. As a result, 83.9% of the needs were judged to have been correctly understood out of the 62 dialogues that reached the final confirmation. Notably, P5’s low accuracy was mainly due to frequent speech recognition errors.

We used G-Eval to assess whether the LLM accurately converted users’ utterances into needs [42]. G-Eval is a tool that uses ab LLM to evaluate natural language generation (NLG) tasks, scoring them from 0 to 1 based on specific criteria. It is not necessary for the LLM model used for the NLG task to match the one used for evaluation in G-Eval; the higher the performance of the evaluation model, the more reliable the assessment. Therefore, gpt-4o model, which outperforms gpt-3.5-turbo, was employed for evaluation [43]. Because user utterances can be affected by microphone accuracy, speech recognition, and colloquial language, we compared the LLM-generated needs (without noise) to the actual spoken utterances to assess noise reduction. The evaluation prompt was “Evaluate whether the user’s request has been accurately and truthfully converted into a structured 6W1H needs format. The 6W1H structure is represented by where, when, who, whom, why, what, and how. The input consists of the user’s request statement and a list of potential service names that might be included in the request. It is acceptable for any 6W1H elements that are not present in the request text to be left blank in the output”. The number of needs utterances evaluated was 100 for the LLM-generated ones and 59 for the actual utterances by participants. The box plot of the resulting scores is shown in Figure 7.

Figure 8 summarizes the time required for each needs detection and Table 2 summarizes the number of responses to the VA’s needs clarification dialogues per participant. The final confirmation is not included. The average detection time for all dialogues was 69.8 s.

To evaluate the frequency of needs detections, the intervals between all needs detections were summarized in the box plot shown in Figure 9. The frequency of needs detection refers to how often the system can track and detect time-varying needs. If the next detection occurred within 5 min, it was considered that there was no significant time variation in needs, and such cases were excluded from the frequency evaluation.

Figure 10 summarizes the results of the post-experiment participant survey. In this experiment, the VA was called Mei. The survey questions were:Q1: Do you frequently use YouTube?Q2: Do you frequently use smart home services?Q3: Do you frequently use SNSs?Q4: Do you often have complaints or requests regarding services?Q5: Were you able to effectively communicate your requests to Mei?Q6: Did you feel that Mei accurately understood your requests?Q7: Did you feel that Mei accurately understood which service you wanted to use?Q8: Did you find Mei’s feature of asking for additional requests useful for accurately communicating your needs?Q9: Was the length of the needs elicitation dialogue appropriate?Q10: Was the dialogue for needs extraction with Mei enjoyable at the beginning of the experiment?Q11: Was the dialogue for needs extraction with Mei consistently enjoyable throughout the experiment?

Participants responded on a five-point Likert scale, where one means strong disagreement and five means strong agreement [44]. Questions Q1 through Q4 focused on understanding the participant characteristics. Question Q5 assessed the practicality of Needs Companion, Q6 and Q7 its reliability, Q8 its effectiveness, Q9 its efficiency, and Q10 and Q11 its pleasantness. These survey items were defined according to the SQuaRE software quality evaluation standards [45].

The number of needs per service for each participant and the survey results regarding service usage and requests are summarized in Figure 11 and Table 3, respectively.

### 7.3. Baseline Experiment

This study is the first to propose extracting needs using the 6W1H format, meaning no prior research exists on human-based 6W1H extraction. To address this, we conducted an interview experiment to compare the performance of Needs Companion with human interviews. In the experiment, one person acted as the interviewer, and the other as the user. The interviewer gathered the user’s service-related needs and organized them into the 6W1H format. Afterward, the interviewer asked the user to confirm whether the needs were accurately summarized. To reduce bias, three software students, who did not participate in the main experiment, conducted the interviews. Three participants from the main experiment (P5, P6, and P7) took on the role of users. Each pair assessed needs for three services: YouTube, smart home services, and SNSs. Nine total assessments were completed. In eight cases, users confirmed the needs as correct, while one case received a “not my need” response. Figure 12 shows the time spent on each assessment, with the average being 237.6 s.

### 7.4. Discussions

The performance of Needs Companion will be discussed in terms of sensor performance based on experimental data and software quality based on questionnaires. The sensor performance of Needs Companion will be discussed in terms of accuracy, response time, and noise reduction. The software quality will be discussed from the viewpoint of the quality at the time of use, and whether it is easy for users to use or not will be clarified.

First, accuracy measures how accurately the needs were detected. The accuracy of the externalized needs data was judged by the users themselves. As shown in Figure 6, needs were accurately detected in 83.9% of all dialogues. Table 4 presents the 95% confidence interval for detection accuracy, calculated using the bootstrap method. The relative width of the confidence interval is 33.8%, which is reasonably small. Given that this is a social science study involving human participants, an accuracy rate of 83.9% is a meaningful result. For comparison, the baseline experiment achieved an extraction accuracy of 88.9%. These results suggest that Needs Companion offers practical accuracy.

Next, response speed was evaluated for two aspects: needs detection time and interval. The average duration for a single needs detection was 69.8 s, compared to 237.6 s in the baseline experiment. As the average detection time did not follow a normal distribution, the Mann–Whitney U test was employed to determine if Needs Companion’s detection time was significantly shorter than that of the interviews. The *p*-value was $1.94\times 10^{-6}$, indicating a statistically significant difference. Therefore, Needs Companion enables faster needs extraction. In the baseline experiment, the interviewer took considerable time to interpret the user’s statements and summarize needs, whereas the LLM’s advanced natural language processing likely contributed to the shorter detection time.

Next, response speed was assessed by how frequently users’ internal needs could be externalized. As indicated in Figure 11, multiple needs for a single service were often detected, suggesting that continuous detection is important as needs may change over time. Table 3 shows that while all participants frequently used YouTube and had many needs detections, multiple needs were also detected for less frequently used services, such as smart home services, suggesting that needs for various services can change over time regardless of usage frequency. Figure 9 and Table 4 show that needs could be extracted from the same user approximately once every one to two weeks, indicating the potential to track weekly changes in internal needs.

Finally, noise reduction was assessed by the system’s ability to detect needs without interference from microphone issues, speech recognition errors, or disorganized input. From Figure 7, we can see that noise reduces the mean of the output accuracy score from 0.80 to 0.56. Figure 6 shows that there exist cases where the accuracy drops considerably due to significantly worse speech recognition accuracy for some subjects. These results indicate that Needs Companion is greatly affected by noise. Thus, while Needs Companion has sufficient accuracy and high response speed as a sensor, it is necessary to take countermeasures against noise.

Based on the questionnaire results shown in Figure 10, we discuss the software quality of Needs Companion. First, the results were found to be excellent with respect to effectiveness and pleasantness. Regarding effectiveness, Q8 revealed that follow-up dialogues were useful for detecting needs. Pleasantness was found to be a continuing motivation to use Needs Companion and observe needs continuously throughout the five-week experiment. When participants were interviewed about the reasons for their high pleasantness scores, they mentioned that they were comfortable interacting with VAs and that the creative dialogue with LLMs was stimulating and never boring. Furthermore, while many participants rated efficiency, practicality, and reliability highly, room for improvement remains in these areas. In terms of efficiency, although the length of each needs detection dialogue was generally received positively, the prevalence of “somewhat positive” ratings suggests there is a need to further reduce dialogue duration. For practicality, while many participants gave positive feedback on Q5, which asked if they could effectively communicate their needs to Needs Companion, the fact that three participants gave a “neutral” rating indicates the system could be improved to make it easier for users to express their needs. Comparing the results of Q6 and Q7, it was found that while many participants answered that the service name itself was adequately observed, others thought that there was still room for improvement in the observation of needs. From the above, it was found that while Needs Companion is software that gives users a high level of satisfaction when they use it, there are several areas for improvement.

### 7.5. Advantages & Limitations

Needs Companion is a needs sensor capable of detecting user-specific service needs with sufficient accuracy and high response speed. Additionally, it can be said that Needs Companion offers high resolution. Here, resolution refers to how effectively Needs Companion’s detections, using the 6W1H needs data model, capture the details of user needs. The needs extraction by Needs Companion, facilitated by an LLM, is conducted with exceptionally high expressive power. As long as the prompts are properly specified, there is minimal risk that different user utterances during the extraction process will be incorrectly converted into the same need. Since the 6W1H needs data model allows each element to be described in natural language, it provides flexibility in representing variations in needs. Furthermore, the fact that each element can express autonomy, competence, and relatedness in terms of BPNs contributes to its high resolution. In addition to these advanced sensor capabilities, the fact that participants were able to use Needs Companion after only brief training demonstrates its realization as a non-specialized and automated sensor. From these detections, it can be concluded that Needs Companion is an innovative and highly effective needs sensor.

The detected needs can be utilized for detailed service personalization through natural language analysis and machine learning. The 6W1H needs data model is described in natural language, making it easily understandable for anyone. We believe that this study’s success in presenting user needs in an easily interpretable format contributes to addressing the critical issues of explainability and interpretability in complex AI applications. Needs Companion also exhibits high software quality. It combines a VA capable of rich interaction through voice, facial expressions, and gestures with the high creativity of an LLM, ensuring continuous and engaging needs detection. This interactive approach, which keeps the user at the center of needs expression, preserves the crucial elements of autonomy and competence, making it a groundbreaking method for needs detection.

Needs Companion has several limitations. Regarding reproducibility, the LLM can be configured to produce more deterministic outputs by adjusting settings such as temperature. However, since LLM models are frequently updated, using models that are less frequently updated and offer greater transparency, such as publicly available models, will be necessary to enhance reproducibility. To improve resolution and reduce noise, it is essential to adopt a more advanced speech-to-text AI to enhance the system’s voice recognition capabilities. Additionally, improving the LLM’s ability to analyze spoken language will require not only advancements in model performance but also further refinement of dialogues and prompts. Specifically, responses could be designed to clarify ambiguous points more effectively, and user-specific dialogue history and needs records could be incorporated as inputs to better understand individual speaking habits. This would allow the LLM to extract needs with a more contextual understanding, leading to the extraction of more personalized, latent needs. One limitation of the evaluation is that the experiments were conducted with a limited group of participants, namely students and faculty members within the laboratory. It remains unclear whether the system can accurately capture the needs of elderly individuals or children, who may not be as familiar with digital services. Moreover, the potential influence of social biases, such as those related to age or disability, in the LLM’s needs extraction process has yet to be evaluated. While the resolution of needs in terms of language is very high, human expression is not limited to language alone. It is also necessary to detect users’ service needs through additional information such as voice pitch and volume, facial expressions, gestures, and interactive timing during dialogues [46].

## 8. Conclusions

The objective of this study is to propose, develop, and evaluate the performance of Needs Companion, a system for automatically detecting individual user service needs. The main idea is to obtain easily interpretable and structured 6W1H needs data through needs elicitation dialogues with a VA and analysis by an LLM. This approach successfully creates a sensor that automatically detects service needs with sufficient accuracy and fast response times, without requiring expert involvement. However, the proposed method faces challenges with noise reduction, requiring improvements in speech recognition and natural language processing. Future research will focus on evaluating rapidly advancing AI technologies to identify and integrate the most suitable ones for enhancing sensor performance. By incorporating AI technologies such as image recognition, Needs Companion will evolve into a multimodal AI sensor. Additionally, the evaluation will be expanded to include elderly individuals, and efforts will be made to develop a bias-free needs sensor. Moreover, the study will explore methods for service personalization by leveraging the detected needs, including machine learning-based recommendations and requirements analysis for service development.

## Figures and Tables

**Figure 1 sensors-24-06814-f001:**
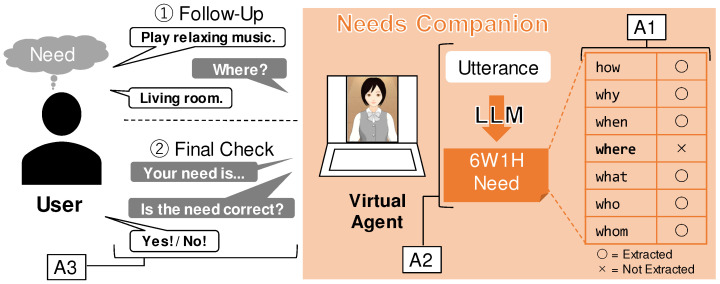
Overall architecture of needs companion.

**Figure 2 sensors-24-06814-f002:**
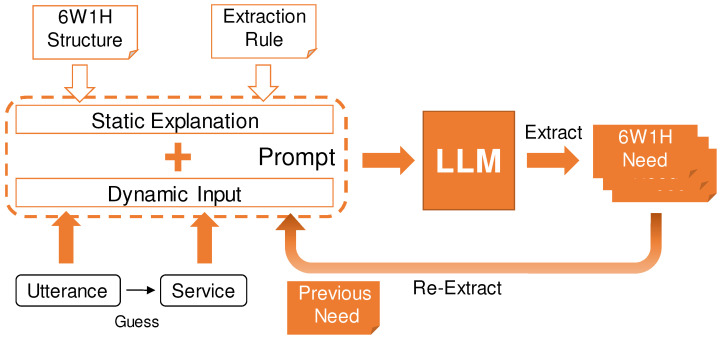
Process of needs extraction using LLM.

**Figure 3 sensors-24-06814-f003:**
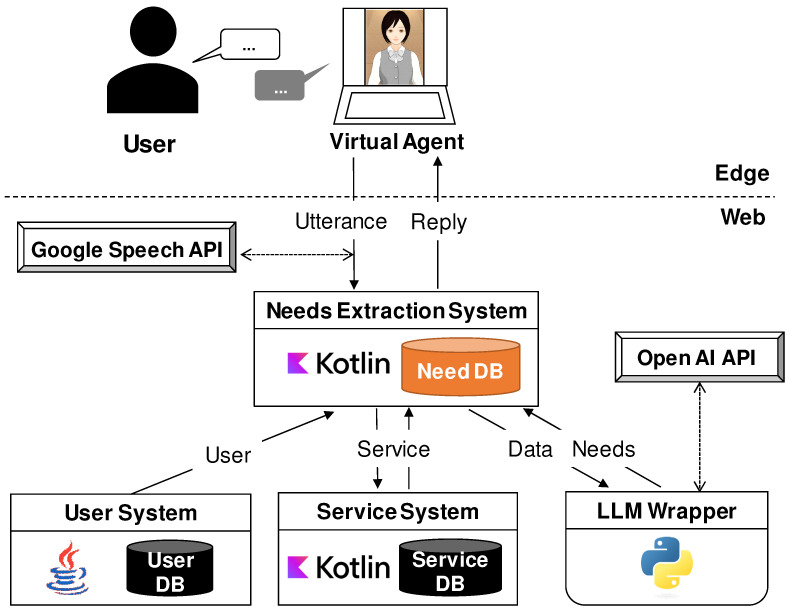
Implementation architecture.

**Figure 4 sensors-24-06814-f004:**
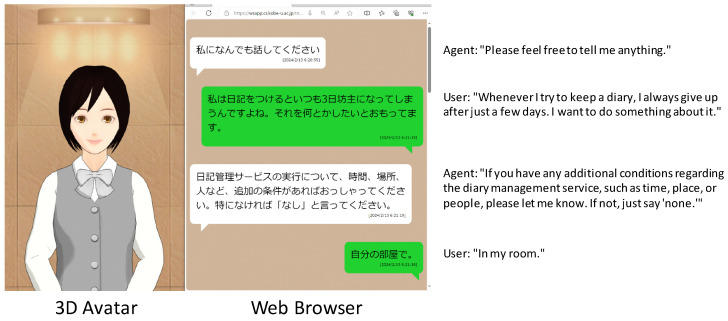
Dialogue screen.

**Figure 5 sensors-24-06814-f005:**
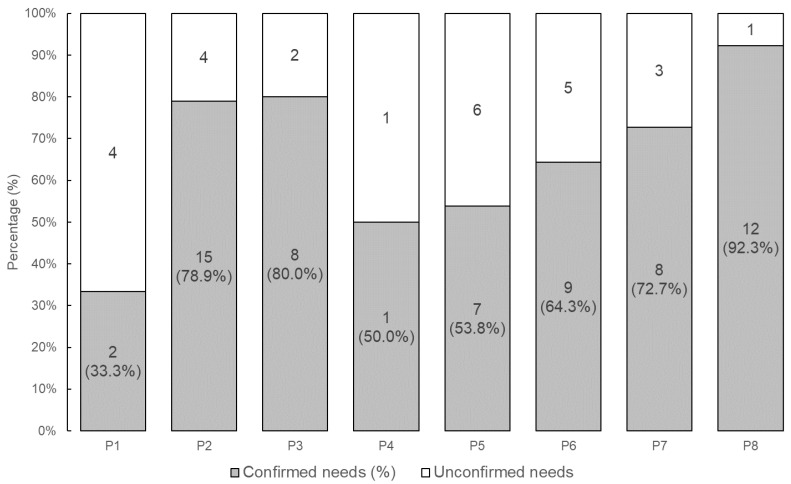
Percentage of needs confirmed by participants.

**Figure 6 sensors-24-06814-f006:**
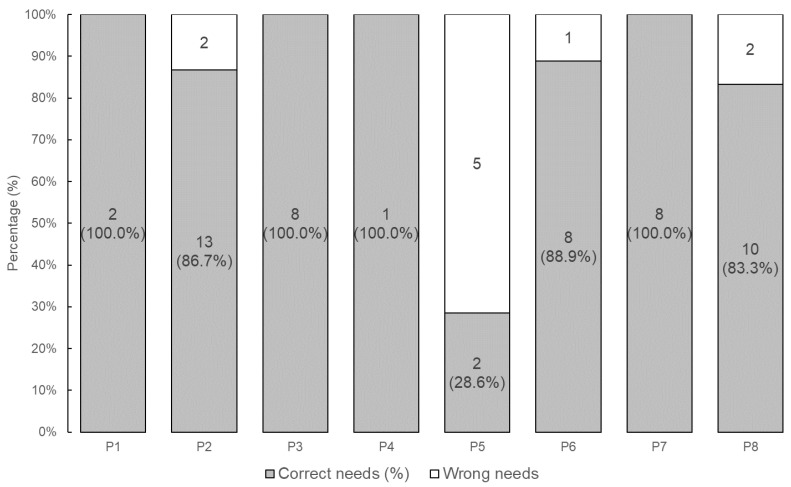
Results of needs confirmation by participants.

**Figure 7 sensors-24-06814-f007:**
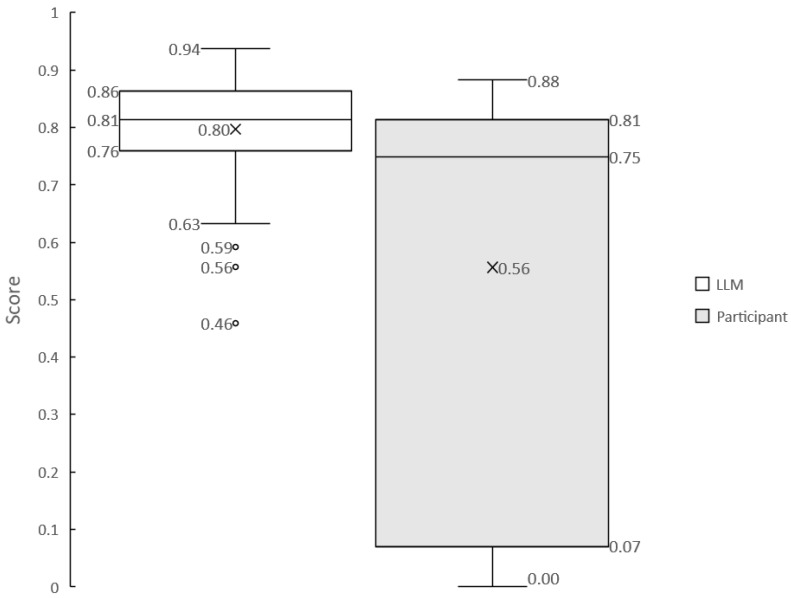
Needs conversion scores by LLM.

**Figure 8 sensors-24-06814-f008:**
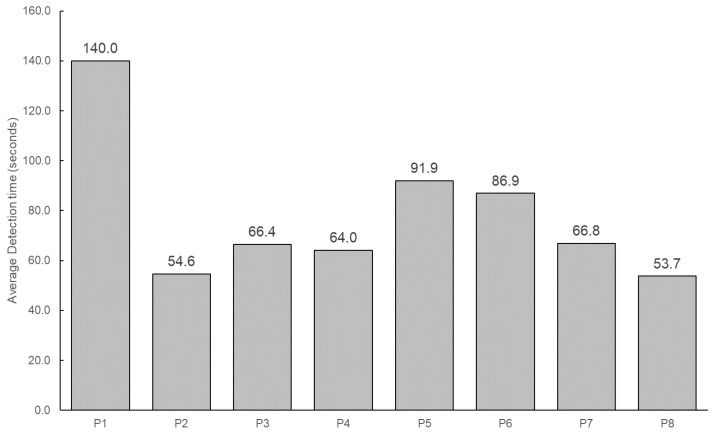
Average needs detection time (seconds).

**Figure 9 sensors-24-06814-f009:**
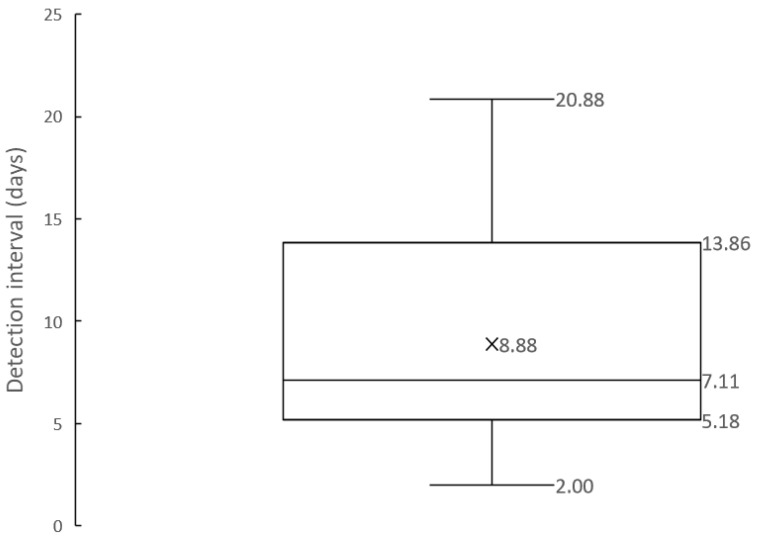
Needs detection intervals (days).

**Figure 10 sensors-24-06814-f010:**
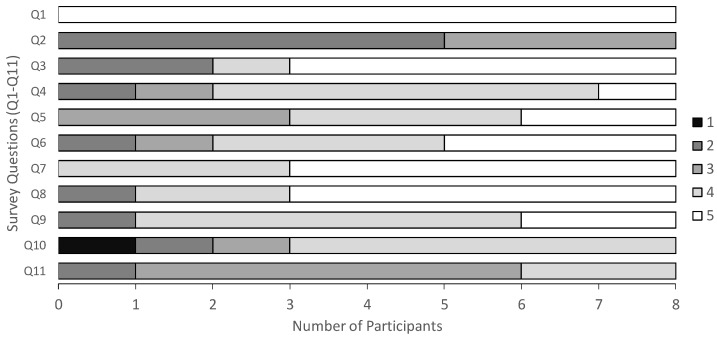
Survey results on a 5-point Likert scale.

**Figure 11 sensors-24-06814-f011:**
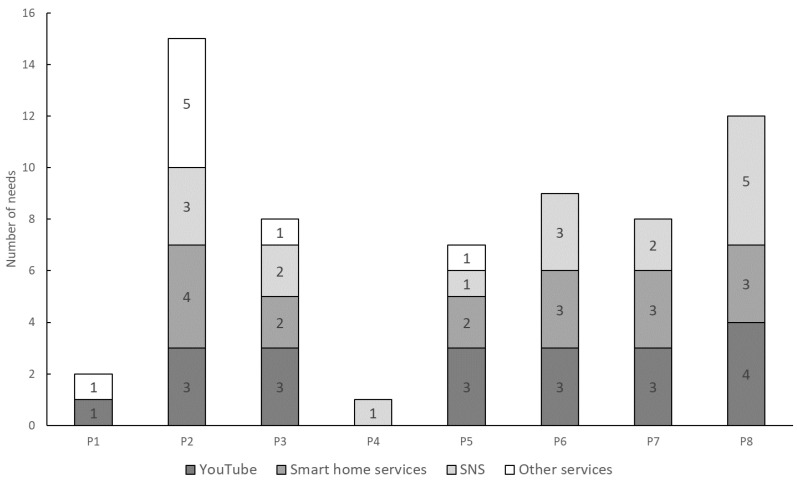
Number of needs per service.

**Figure 12 sensors-24-06814-f012:**
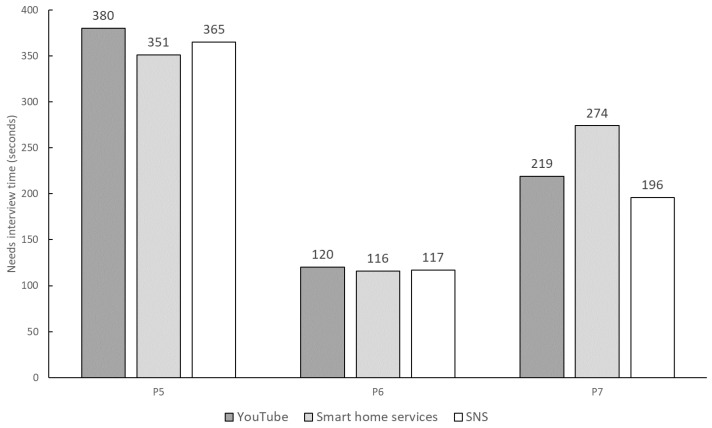
Time spent on needs assessment interviews (seconds).

**Table 1 sensors-24-06814-t001:** 6W1H service needs data model.

6W1H	Type	Content
how	Service	What service do you want to use?
why	Autonomy	Why do you want to do it?
when	Autonomy	When do you want to do it?
where	Autonomy	Where do you want to do it?
what	Competence	What exactly do you want to do with the service?
who	Relatedness	Who is the main actor?
whom	Relatedness	To whom do you want to do it?

**Table 2 sensors-24-06814-t002:** Extent to which participants replied to the needs clarification dialogue.

Participant	Median Number of Replies	Reply Rate
P1	4.5	100%
P2	0	6.7%
P3	1	87.5%
P4	1	100%
P5	1	100%
P6	1	88.9%
P7	1	62.5%
P8	0	30.8%

**Table 3 sensors-24-06814-t003:** Survey results on service usage and service requests based on a 5-point Likert scale.

Participant	Q1 (YouTube Usage)	Q2 (Smart Home Services Usage)	Q3 (SNS Usage)	Q4 (Requests)
P1	5	3	2	3
P2	5	3	2	5
P3	5	2	5	4
P4	5	2	5	4
P5	5	3	5	4
P6	5	2	5	2
P7	5	2	4	4
P8	5	2	5	4

**Table 4 sensors-24-06814-t004:** The 95% confidence intervals using the bootstrap method.

Value	Average	Lower Bound	Upper Bound	Relative Width	Sample Size
Accuracy (%)	83.9	67.2	95.6	33.8%	62
Detection Time (seconds)	69.8	62.1	77.8	22.5%	62
Detection Interval (days)	8.88	6.70	11.4	53.0%	20

## Data Availability

The data presented in this study are available on request from the corresponding author. The data are not publicly available due to privacy restrictions.

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
