# Peer review of "Needs Companion: A Novel Approach to Continuous User Needs Sensing Using Virtual Agents and Large Language Models"

_sensors, 2024, doi:10.3390/s24216814_

Round 1

Reviewer 1 Report

Comments and Suggestions for Authors

This article presents a novel approach to continuous user needs sensing by integrating virtual agents with large language models. The use of the 6W1H framework, coupled with the experimental validation, is highly commendable. However, several aspects of the article would benefit from further clarification and improvement.

General Concept Comments:

Please add real-world examples or screenshots from the actual implementation of the Needs Companion system. This would provide clearer insights into its practical applications and enhance the reader's understanding of the system's functionality.

The evaluation section appears somewhat disorganized. The main issues include an overly broad scope, redundant discussions, and a conflation of user survey results with performance evaluation metrics. I recommend restructuring this section by separating different types of evaluations and focusing discussions on key findings and insights to avoid repetition. 

Additionally, to validate the performance of the "Needs Companion" system more comprehensively, it would be worthwhile to consider conducting tests with participants from diverse backgrounds and age groups. If expanding the sample size and diversity is not feasible, I suggest acknowledging this limitation in the discussion section.

In the implementation part of the paper, it is mentioned that gpt-3.5-turbo was employed for needs extraction. However, in the evaluation section, GPT-4o is used for G-Eval evaluation. Please clarify why different models were used in the implementation and evaluation stages. Were there specific reasons or considerations behind this choice? If feasible, using the same model for both implementation and evaluation would enhance the consistency and reliability of the results.

The discussion of limitations could be expanded, particularly regarding potential biases inherent in the LLM and virtual agents. 

Specific Comments:

Figure 2 illustrates the process of needs extraction using LLM, but the "Re-extract" arrow pointing to the LLM does not clearly show how the previously extracted data is fed back into the model. Furthermore, the representation of the 6W1H structure as prompts for the LLM is somewhat ambiguous and may cause confusion. More clarity in this aspect would be helpful.

Section 5.5, Lines 164-166 lacks specific citations to support the claims made. Providing relevant references here would strengthen the arguments.

Section 5.5, Lines 174-176 contains some assumptions and speculations. Such statements should either be supported by concrete research or framed as hypotheses or theoretical speculations.

Figure 4 shows a clear distinction between systems, but a slight reorganization could better demonstrate the interaction flow. For instance, using more directional arrows or clearer labeling could help clarify the data flow, especially between the "Needs Extraction System" and other components.

Section 7.4, Lines 318-320 mentions the VA's abilities related to facial expressions and gestures, which were not previously discussed in the paper. It would be beneficial to revise this section for consistency and clarify the relevance of these capabilities.

Comments on the Quality of English Language

The English language in the manuscript is generally clear, but minor editing is needed to improve readability and clarity, particularly by simplifying complex sentences and ensuring consistent use of terminology throughout the manuscript. 

Reviewer 2 Report

Comments and Suggestions for Authors

The topic of users need sensing using virtual agents and llms is of interest and quite relevant. The authors have proposed a strategy to convert users needs into structure database following 6W1H format. I have some concerns as follows:
1. In the current explanation and experiment provided, how it can be differentiated from a conventional virtual assistant? 

2. The system is not providing any personalized recommendations as no contextual or previous queries (needs/dialogues) are incorporated. I think that could be the real contribution of this study.

3. The sample size is too small to validate the proposed methodology. Only 8 subjects with max of 15 confirmed needs and min of only 1 confirmed need. The subject with only 1 confirmed request resulting in 100% correct rate, is inducing bias to the overall test accuracy of 83.3%. Same is the case with most of the subjects. It is suggested to increase the sample size to properly validate the results.

4. No comparison is provided with other existing state-of-the art methods. 

5. Overall, the authors need major revisions to enhance the experimental design, contextual information and validation strategies. 

Comments on the Quality of English Language

English is well written, overall.
Line 36-38 is written in future tense (will be), which I believe was not intended.

Reviewer 3 Report

Comments and Suggestions for Authors

This paper introduces Needs Companion, a system that uses virtual agents and LLMs to observe user needs in real-time, structured through the 6W1H framework. The approach is leveraging natural language dialogues to extract needs automatically with an accuracy rate of 83.3%.

Strengths:

1. Innovative integration of LLMs and virtual agents for real-time needs sensing.

2. Clear methodology with structured needs extraction.

3. User-centered design that enhances personalization.

One of my concern is the accuracy of extraction: 83.3%.  I am not confident if the accuracy is good or not. Can the author discuss more about the accuracy, how you exaluate it compared to the baseline models? It is missing in the evaluation section.

Comments on the Quality of English Language

The english is clear.

Round 2

Reviewer 1 Report

Comments and Suggestions for Authors

The revised manuscript currently has no major issues. Figures 1 and 3 appear quite similar in form, and I suggest merging them to avoid redundancy and clutter. Additionally, the tables in Section 7 would be better presented as figures, which could provide a clearer and more intuitive view of the results for readers.
